# Equivariant Manifold Flows

**Isay Katsman** [* 1]   **Aaron Lou** [* 1]   **Derek Lim** [* 1]   **Qingxuan Jiang** [* 1]   **Ser-nam Lim** [2]   **Christopher de Sa** [1]

## Abstract

Tractably modelling distributions over manifolds has long been an important goal in the natural sciences. Recent work has focused on developing general machine learning models to learn such distributions. However, for many applications these distributions must respect manifold symmetries— a trait which most previous models disregard. In this paper, we lay the theoretical foundations for learning symmetry-invariant distributions on arbitrary manifolds via equivariant manifold flows. We demonstrate the utility of our approach by using it to learn gauge invariant densities over $SU(n)$ in the context of quantum field theory.

## 1. Introduction

Density learning over manifolds has a broad array of applications ranging from quantum field theory in physics (Wirnsberger et al., 2020) to motion estimation in robotics (Feiten et al., 2013) to protein-structure prediction in computational biology (Hamelryck et al., 2006). Recent work (Lou et al., 2020; Mathieu & Nickel, 2020; Falorsi & Forré, 2020) has extended the powerful framework of continuous normalizing flows (Chen et al., 2018; Grathwohl et al., 2019) to the setting of Riemannian manifolds, lifting the utility of these models for learning complex probability distributions to a more general setting.

Although these manifold normalizing flows were a considerable step forward, they are insufficient for many problems in the natural sciences. For example, coupled particle systems in physical chemistry (Köhler et al., 2020) and $SU(n)$ lattice gauge theories in theoretical physics (Boyda et al., 2020) require symmetries that are nontrivial to enforce. Typically, these symmetries are enforced in an ad hoc way using properties specific to the manifold (Boyda et al., 2020). In

---
[*]Equal contribution  [1]Department of Computer Science, Cornell University, Ithaca, New York, USA [2]Facebook AI, New York, New York, USA. Correspondence to: Isay Katsman <isk22@cornell.edu>.

Third workshop on *Invertible Neural Networks, Normalizing Flows, and Explicit Likelihood Models* (ICML 2021). Copyright 2021 by the author(s).

contrast, our paper presents a fully general way to learn flows that induce symmetry-invariant distributions.

## 2. Related Work

**Normalizing Flows on Manifolds** Normalizing flows on manifolds have received a considerable amount of attention, both in terms of manifold-specific and general constructions. Rezende et al. (2020) introduced constructions specific to tori and spheres, while Bose et al. (2020) introduced constructions for hyperbolic space. Following this work, Lou et al. (2020); Mathieu & Nickel (2020) introduced a general construction by extending Neural ODEs (Chen et al., 2018) to the setting of Riemannian manifolds.

**Equivariant Machine Learning** Recent work has incorporated equivariance into machine learning models for the purpose of modelling symmetries (Cohen & Welling, 2016; Kondor & Trivedi, 2018; Rezende et al., 2019). In particular, Köhler et al. (2020) introduced equivariant normalizing flows for Euclidean space and Boyda et al. (2020) introduced equivariant flows for $SU(n)$ via a manifold-specific construction. In contrast, the equivariant manifold flows in our paper are fully general and applicable to arbitrary Riemannian manifolds.

## 3. Background

In this section, we provide a terse overview of the necessary concepts for understanding our paper. For a more detailed introduction to Riemannian geometry, we refer the reader to texts such as Lee (2013); Kobyzev et al. (2020).

### 3.1. Riemannian Geometry

A Riemannian manifold $(\mathcal{M}, h)$ is an $n$-dimensional manifold with a smooth collection of inner products $(h_x)_{x \in \mathcal{M}}$ for every tangent space $T_x \mathcal{M}$. The Riemannian metric $h$ induces a distance $d_h$ on the manifold.

A diffeomorphism $f : \mathcal{M} \to \mathcal{M}$ is called an isometry if $h(D_x f(u), D_x f(v)) = h(u, v)$ for all tangent vectors $u, v \in T_x \mathcal{M}$ where $D_x f$ is the differential of $f$. Note that isometries preserve the manifold distance function. The collection of all isometries forms a group $G$, which we call the isometry group of the manifold $\mathcal{M}$.

Riemannian metrics also allow for a natural analogue of gradients on $\mathbb{R}^n$. For a function $f : \mathcal{M} \to \mathbb{R}$, we define the Riemannian gradient $\nabla_x f$ to be the vector on $T_x \mathcal{M}$ such that $h(\nabla_x f, v) = D_x f(v)$ for $v \in T_x \mathcal{M}$.

### 3.2. Normalizing Flows on Manifolds

Let $(\mathcal{M}, h)$ be a Riemannian manifold. A normalizing flow on $\mathcal{M}$ is a diffeomorphism $f_\theta : \mathcal{M} \to \mathcal{M}$ (parametrized by $\theta$) that transforms a prior density $\rho$ to model density $\rho_{f_\theta}$. The model distribution can be computed via the change of variables equation:

$$\rho_{f_\theta}(x) = \rho \left( f_\theta^{-1}(x) \right) \left| \det \frac{\partial f_\theta^{-1}(x)}{dx} \right|.$$

### 3.3. Equivariance and Invariance

Let $G$ be an isometry subgroup of $\mathcal{M}$. We notate the action of an element $g \in G$ on $\mathcal{M}$ by the map $R_g : \mathcal{M} \to \mathcal{M}$.

**Equivariant and Invariant Functions**   We say that a function $f : \mathcal{M} \to \mathcal{N}$ is equivariant if, for symmetries $g_x : \mathcal{M} \to \mathcal{M}$ and $g_y : \mathcal{N} \to \mathcal{N}$, $f \circ g_x = g_y \circ f$. We say a function $f : \mathcal{M} \to \mathcal{N}$ is invariant if $f \circ g_x = f$. When $X$ and $Y$ are manifolds, the symmetries $g_x$ and $g_y$ are isometries.

**Equivariant Vector Fields**   Let $X : \mathcal{M} \times [0, \infty) \to T\mathcal{M}$, $X(m, t) \in T_m \mathcal{M}$ be a time-dependent vector field on manifold $\mathcal{M}$, with base point $x_0 \in \mathcal{M}$. $X$ is a $G$-equivariant vector field if $\forall (m, t) \in \mathcal{M} \times [0, \infty)$, $X(R_g m, t) = (D_m R_g) X(m, t)$.

**Equivariant Flows**   A flow $f$ on a manifold $\mathcal{M}$ is $G$-equivariant if it commutes with actions from $G$, i.e. we have $R_g \circ f = f \circ R_g$.

**Invariance of Density**   For a group $G$, a density $\rho$ on a manifold $\mathcal{M}$ is $G$-invariant if, for all $g \in G$ and $x \in \mathcal{M}$, $\rho(R_g x) = \rho(x)$, where $R_g$ is the action of $g$ on $x$.

## 4. Invariant Densities from Equivariant Flows

In this section, we describe a tractable way to learn a density over a manifold that obeys a symmetry given by an isometry subgroup $G$. Directly parameterizing a density that obeys this symmetry is nontrivial. Hence we prove the following implications that yield a tractable solution to this problem (note that we generalize previous work (Köhler et al., 2020; Papamakarios et al., 2019) that has only addressed the case of Euclidean space):

1. **G-invariant potential $\Rightarrow$ G-equivariant vector field (Theorem 1).** We show that given a $G$-invariant potential function $f : \mathcal{M} \to \mathbb{R}$, the vector field $\nabla f$ is $G$-equivariant.

2. **G-equivariant vector field $\Rightarrow$ G-equivariant flow (Theorem 2).** We show that a $G$-equivariant vector field on $\mathcal{M}$ uniquely induces a $G$-equivariant flow.

3. **G-equivariant flow $\Rightarrow$ G-invariant density (Theorem 3).** We show that given a $G$-invariant prior $\rho$ and a $G$-equivariant flow $f_\theta$, the flow density $\rho_{f_\theta}$ is $G$-invariant.

Hence, we can obtain a $G$-invariant density from a $G$-invariant potential. We claim that constructing a $G$-invariant potential function on a manifold is far simpler than directly parameterizing a $G$-invariant density or a $G$-equivariant flow (an example construction will be given). We defer the proofs of all theorems to Appendix A.

### 4.1. Equivariant Gradient of Potential Function

We start by showing how to construct $G$-equivariant vector fields from $G$-invariant potential functions. To design an equivariant vector field $X$, it is sufficient to set the vector field dynamics of $X$ as the gradient of some $G$-invariant potential function $\Phi : \mathcal{M} \to \mathbb{R}$:

**Theorem 1.** *Let $(\mathcal{M}, h)$ be a Riemannian manifold and $G$ be its group of isometries (or an isometry subgroup). If $\Phi : \mathcal{M} \to \mathbb{R}$ is a smooth $G$-invariant function, then the following diagram commutes for any $g \in G$:*

$$
\begin{array}{ccc}
\mathcal{M} & \xrightarrow{R_g} & \mathcal{M} \\
\downarrow{\scriptstyle \nabla\Phi} & & \downarrow{\scriptstyle \nabla\Phi} \\
T\mathcal{M} & \xrightarrow{DR_g} & T\mathcal{M}
\end{array}
$$

*or $\nabla_{R_g u}\Phi = D_u R_g(\nabla_u \Phi)$. Hence $\nabla \Phi$ is a $G$-equivariant vector field. This condition is also tight in the sense that it only occurs if $G$ is the group of isometries.*

### 4.2. Constructing Equivariant Manifold Flows from Equivariant Vector Fields

To construct equivariant manifold flows, we use tools from manifold ordinary differential equations (ODEs) and continuous normalizing flows (CNFs). In particular, note the definition below:

**Manifold Continuous Normalizing Flows** A manifold continuous normalizing flow with base point $z$ is a function $\gamma : [0, \infty) \to \mathcal{M}$ that satisfies the manifold ODE

$$\frac{d\gamma(t)}{dt} = X(\gamma(t), t), \gamma(0) = z.$$

We define $F_{X,T} : \mathcal{M} \to \mathcal{M}$, $z \mapsto F_{X,T}(z)$ to map any base point $z \in \mathcal{M}$ to the value of the CNF starting at $z$, evaluated at time $T$. This function is known as the (vector field) flow of $X$. Observe that there exists a natural correspondence

between equivariant flows and equivariant vector fields, by the following theorem:

**Theorem 2.** *Let $(\mathcal{M}, h)$ be a Riemannian manifold, and $G$ be its isometry group (or one of its subgroups). Let $X$ be any time-dependent vector field on $\mathcal{M}$, and $F_{X,T}$ be the flow of $X$. Then $X$ is a $G$-equivariant vector field if and only if $F_{X,T}$ is a $G$-equivariant flow.*

### 4.3. Invariant Manifold Densities from Equivariant Flows

We now show that $G$-equivariant flows induce $G$-invariant densities. Note that we require the group $G$ to be an isometry subgroup in order to control the density of $\rho_f$, and the following theorem does not hold for general diffeomorphism groups.

**Theorem 3.** *Let $(\mathcal{M}, h)$ be a Riemannian manifold, and $G$ be its isometry group (or one of its subgroups). If $\rho$ is a $G$-invariant density on $\mathcal{M}$, and $f$ is a $G$-equivariant diffeomorphism, then $\rho_f(x)$ is also $G$-invariant.*

### 4.4. Universality of Flows Generated by Invariant Potentials

We prove that flows induced by invariant potentials suffice to learn any smooth invariant distribution over a closed manifold, as measured by Kullback-Leibler (KL) divergence.

**Theorem 4.** *Let $(\mathcal{M}, h)$ be a closed Riemannian manifold. Let $\rho, \pi$ be smooth $G$-invariant distributions over said manifold, and let $D_{KL}(\rho||\pi)$ be the KL divergence between distributions $\rho$ and $\pi$. If we choose a function $g : \mathcal{M} \to \mathbb{R}$ such that for $x \in \mathcal{M}$,*

$$g(x) = \log\left(\frac{\pi(x)}{\rho(x)}\right).$$

*Then we have:*

$$\frac{\partial}{\partial t} D_{KL}(\rho||\pi) = -\int_{\mathcal{M}} \rho \exp(g)\|\nabla g\|^2 \, dx \le 0.$$

In particular, note that if the target distribution is $\pi$ and the current distribution is $\rho$, if we set $g$ to be $\log(\pi(x)/\rho(x))$ and $g$ is the potential from which the flow is obtained, then the KL divergence between $\pi$ and $\rho$ is monotonically decreasing by Theorem 4.

## 5. Learning Invariant Densities with Equivariant Flows

In this section, we discuss how the theory in Section 4 is applied.

### 5.1. Equivariant Manifold Flow Model

We assume that a $G$-invariant potential function $f : \mathcal{M} \to \mathbb{R}$ is given. The equivariant flow model works by using automatic differentiation (Paszke et al., 2017) on $f$ to obtain $\nabla f$, using this for the vector field, and integrating in a step-wise fashion over the manifold. Specifically, forward integration and change-in-density (divergence) computations utilize the Riemannian CNF (Mathieu & Nickel, 2020) framework. This flow model is used with a specific training procedure (see Section 5.3) to obtain a $G$-invariant model density that approximates some target.

### 5.2. Constructing Conjugation-invariant Potential Functions on $SU(n)$

For many applications in physics (specifically gauge theory and lattice quantum field theory), one works with the Lie group $SU(n)$ — the group of unitary matrices with determinant 1 (for details on the manifold structure of $SU(n)$, see Appendix B). In particular, when modelling probability distributions on $SU(n)$ for lattice QFT, the desired distribution must be invariant under conjugation by $SU(n)$ (Boyda et al., 2020). Conjugation is an isometry on $SU(n)$ (see Appendix A.5), so we can model probability distributions invariant under this action with our developed theory.

**Invariant Potential Parameterization** We want to produce a $G$-invariant potential function $\Phi : SU(n) \to \mathbb{R}$. Note that matrix conjugation preserves eigenvalues. Thus, for a function $\Phi : SU(n) \to \mathbb{R}$ to be invariant to matrix conjugation, it has to act on the eigenvalues of $x \in SU(n)$ as a multi-set.

We can parameterize such potential functions $\Phi$ by the DeepSet network from Zaheer et al. (2017). DeepSet is a permutation invariant neural network that acts on the eigenvalues, so the mapping of $x \in SU(n)$ is $\Phi(x) = \hat{\Phi}(\{\lambda_1(x), \ldots, \lambda_n(x)\})$ for some set function $\hat{\Phi}$.

As a result of this design, the only variance in the learned distribution will be amongst non-similar matrices, while all similar matrices will be assigned the same density value.

**Prior Distributions** For the prior distribution of the flow, we need a distribution that is invariant to conjugation by $SU(n)$. We use the Haar measure, whose volume element is given for $x \in SU(n)$ as $\mathrm{Haar}(x) = \prod_{i<j} |\lambda_i(x) - \lambda_j(x)|^2$ (Boyda et al., 2020). We can sample from and compute log probabilities for this distribution efficiently with standard matrix computations (Mezzadri, 2007).

### 5.3. Training Equivariant Manifold Flows

Learning to sample given an exact density is important in settings such as the one in Boyda et al. (2020), where we are given conjugation-invariant densities on $SU(n)$ for which we know the exact density function yet sampling well

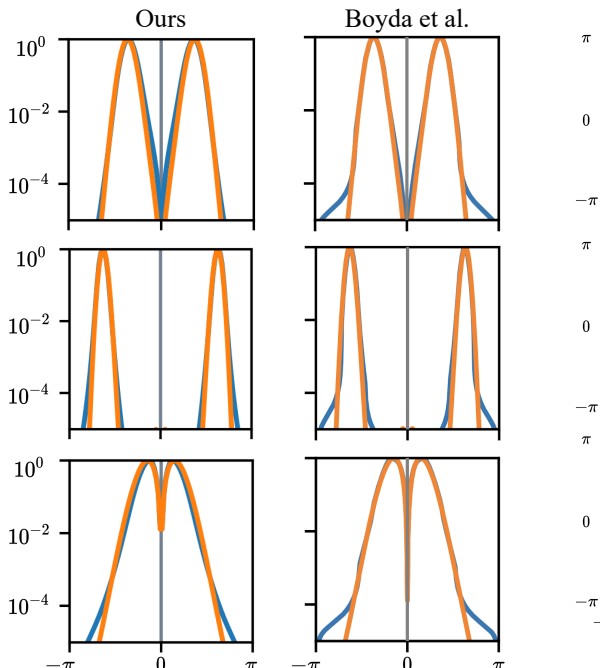

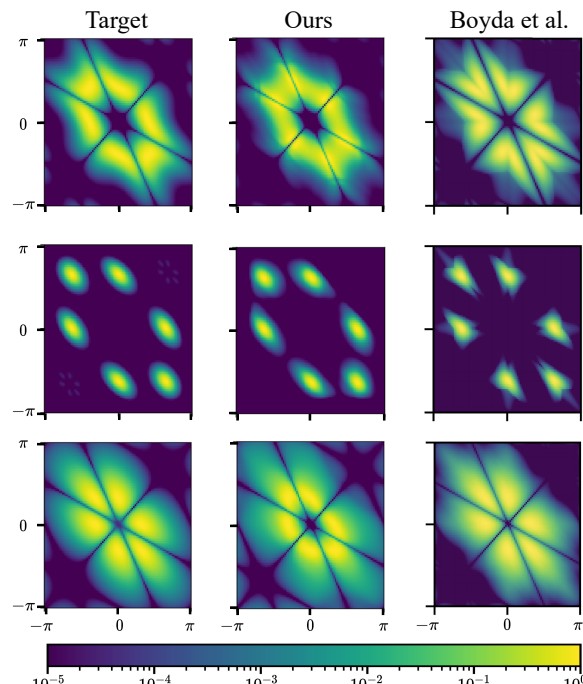

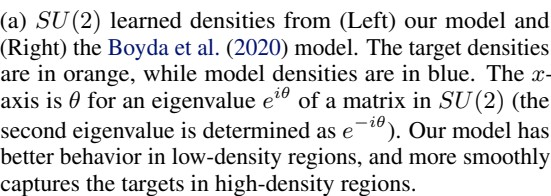

(a) $SU(2)$ learned densities from (Left) our model and (Right) the Boyda et al. (2020) model. The target densities are in orange, while model densities are in blue. The $x$-axis is $\theta$ for an eigenvalue $e^{i\theta}$ of a matrix in $SU(2)$ (the second eigenvalue is determined as $e^{-i\theta}$). Our model has better behavior in low-density regions, and more smoothly captures the targets in high-density regions.

(b) $SU(3)$ learned densities from (Middle) our model and (Right) the Boyda et al. (2020) model for different target densities (Left). The $x$-axis and $y$-axis are the angles $\theta_1$ and $\theta_2$ for eigenvalues $e^{i\theta_1}$ and $e^{i\theta_2}$ of a matrix in $SU(3)$ (the third eigenvalue is determined as $e^{-i\theta_1 - i\theta_2}$), and the probabilities correspond to colors on a logarithmic scale. Our model better captures the geometry of the target densities.

*Figure 1.* Comparison of learned densities on (a) $SU(2)$ and (b) $SU(3)$. All densities are normalized to have maximum value 1.

is nontrivial. We train our models by sampling from the Haar distribution on $SU(n)$, computing the KL divergence between the model probabilities and target probabilities at these samples, and backpropagating from this KL loss.

## 6. Experiments

In this section, we learn densities on $SU(n)$ that are invariant to conjugation by $SU(n)$, which is important for constructing flow-based samplers for $SU(n)$ lattice gauge theories in theoretical physics (Boyda et al., 2020). Our model outperforms the construction given in Boyda et al. (2020). An experiment on an additional manifold with a new symmetry is given in Appendix D. Moreover, we demonstrate in Appendix D that leveraging symmetries inherent to the manifold improves performance over general manifold flows (Lou et al., 2020).

For the sake of staying true to the application area, we follow the framework of Boyda et al. (2020) in learning densities on $SU(n)$ that are invariant to conjugation by $SU(n)$. In particular, our goal is to learn a flow to model a target distribution so that we may efficiently sample from it. As mentioned above in Section 5.3, this setting follows the

paradigm in which we are given exact density functions and learn how to sample. Our model is as described in Section 5; further training details are given in Appendix C.1.

Figure 1a displays learned densities for our model and the model of Boyda et al. (2020) for three densities on $SU(2)$ described in Appendix C.2.1. While both models match the target distributions well in high-density regions, our model exhibits a considerable improvement in lower-density regions, where the tails of our learned distribution decay faster. By contrast, the model of Boyda et al. (2020) seems to be unable to reduce mass near $\pm\pi$, a possible consequence of their construction. Even in high-density regions, our model appears to vary smoothly, with fewer unnecessary bumps and curves compared to the densities of the model in Boyda et al. (2020).

Figure 1b displays learned densities for our model and the model of Boyda et al. (2020) for three densities on $SU(3)$ described in Appendix C.2.2. In this case, our models fit the target densities more accurately and better respect the geometry of the target distribution. Indeed, while the models of Boyda et al. (2020) are often sharp and have pointed corners, our models learn densities that vary smoothly and curve in

ways that are representative of the target distributions.

## 7. Conclusion

In this work, we introduce equivariant manifold flows in a fully general context and provide the necessary theory to ensure a principled construction. We also demonstrate the efficacy of our approach in the context of learning a conjugation invariant density over $SU(n)$, which is an important task for sampling $SU(n)$ lattice gauge theories in quantum field theory.

## Acknowledgements

We would like to thank Facebook AI for funding equipment that made this work possible; in addition, we thank the National Science Foundation for helping fund this research effort (NSF IIS-2008102). We would also like to acknowledge Jonas Köhler and Denis Boyda for their useful insights.

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

# Appendix

## A. Proof of Theorems

In this section, we restate and prove the theorems in Section 4. These give the theoretical foundations that we use to build our models. Prior work (Wasserman, 1969; Field, 1980) addresses some of the results we formalize below.

### A.1. Proof of Theorem 1

**Theorem 1.** *Let $(\mathcal{M}, h)$ be a Riemannian manifold and $G$ be its group of isometries (or an isometry subgroup). If $\Phi : \mathcal{M} \to \mathbb{R}$ is a smooth $G$-invariant function, then the following diagram commutes for any $g \in G$:*

$$
\begin{array}{ccc}
\mathcal{M} & \xrightarrow{R_g} & \mathcal{M} \\
\downarrow{\scriptstyle \nabla\Phi} & & \downarrow{\scriptstyle \nabla\Phi} \\
T\mathcal{M} & \xrightarrow{DR_g} & T\mathcal{M}
\end{array}
$$

*or $\nabla_{R_g u}\Phi = D_u R_g(\nabla_u \Phi)$. This is condition is also tight in the sense that it only occurs if $G$ is the group of isometries.*

*Proof.* We first recall the Riemannian gradient chain rule:

$$\nabla_u(\Phi \circ R_g) = (D_u R_g)^\top (\nabla_{R_g u}\Phi)$$

where $(D_u R_g)^\top : T_{R_g u}\mathcal{M} \to T_u\mathcal{M}$ is the "adjoint" given by

$$h\left(D_u R_g(v), w\right) = h\left(v, (D_u R_g)^\top(w)\right).$$

Since $R_g$ is an isometry, we also have

$$h(x, y) = h\left(D_u R_g(x), D_u R_g(y)\right).$$

Combining the above two equations gives

$$h(x, y) = h(Du R_g(x), D_u R_g(y)) = h\left(x, (D_u R_g)^\top \left(D_u R_g(y)\right)\right),$$

which implies for all $y$,

$$h\left(x, y - (D_u R_g)^\top(D_u R_g(y))\right) = 0.$$

Since $h$ is a Riemannian metric (even pseudo-metric works due to non-degeneracy), we must have that $(D_u R_g)^\top \circ (D_u R_g) = I$.

To complete the proof, we recall that $\Phi = \Phi \circ R_g$, and this combined with chain rule gives

$$\nabla_u \Phi = \nabla_u(\Phi \circ R_g) = (D_u R_g)^\top (\nabla_{R_g u}\Phi).$$

Now applying $D_u R_g$ on both sides gives

$$\nabla_{R_g u}\Phi = D_u R_g \nabla_u \Phi$$

which is exactly what we want to show.

We see that this is an "only if" condition because we must necessarily get that the adjoint is the inverse, which implies that $R_g$ is an isometry. □

## A.2. Proof of Theorem 2

**Theorem 2.** *Let $(\mathcal{M}, h)$ be a Riemannian manifold, and $G$ be its isometry group (or one of its subgroups). Let $X$ be any time-dependent vector field on $\mathcal{M}$, and $F_{X,T}$ be the flow of $X$. Then $X$ is an $G$-equivariant vector field if and only if $F_{X,T}$ is a $G$-equivariant flow for any $T \in [0, +\infty)$.*

*Proof.* **$G$-equivariant $X \Rightarrow G$-equivariant $F_{X,T}$**. We invoke the following lemma from Lee (2013, Corollary 9.14):

**Lemma 1.** *Let $F : \mathcal{M} \to \mathcal{N}$ be a diffeomorphism. If $X$ is a smooth vector field over $\mathcal{M}$ and $\theta$ is the flow of $X$, then the flow of $F_*X$ ($F_*$ is another notation for the differential of $F$) is $\eta_t = F \circ \theta_t \circ F^{-1}$, with domain $N_t = F(M_t)$ for each $t \in \mathbb{R}$.*

Examine $R_g$ and its action on $X$. Since $X$ is $G$-equivariant, we have for any $(x, t) \in \mathcal{M} \times [0, +\infty)$,

$$((R_g)_*X)(x, t) = (D_{R_g^{-1}(x)}R_g)X(R_g^{-1}(x), t) = X(R_g \circ R_g^{-1}(x), t) = X(x, t)$$

so it follows that $(R_g)_*X = X$. Applying the lemma above, we get:

$$F_{(R_g)_*X,T} = R_g \circ F_{X,T} \circ R_g^{-1}$$

and, by simplifying, we get that $F_{X,T} \circ R_g = R_g \circ F_{X,T}$, as desired.

**$G$-equivariant $X \Leftarrow G$-equivariant $F_{X,T}$**. This direction follows from the chain rule. If $F_{X,T}$ is $G$-equivariant, then at all times we have:

$$
\begin{aligned}
(D_m R_g)\left(X(F_{X,t}(m), t)\right) = (D_m R_g)\left(\frac{d}{dt}F_{X,T}(m)\right) && \text{(definition)}\\
= \frac{d}{dt}(R_g \circ F_{X,T})(m) && \text{(chain rule)}\\
= \frac{d}{dt}F_{X,T}(R_g m) && \text{(equivariance)}\\
= X(R_g(F_{X,t}(m)), t) && \text{(definition)}
\end{aligned}
$$

This concludes the proof of the backward direction. □

## A.3. Proof of Theorem 3

**Theorem 3.** *Let $(\mathcal{M}, h)$ be a Riemannian manifold, and $G$ be its isometry group (or one of its subgroups). If $\rho$ is a $G$-invariant density on $\mathcal{M}$, and $f$ is a $G$-equivariant diffeomorphism, then $\rho_f(x)$ is also $G$-invariant.*

*Proof.* We wish to show $\rho_f(x)$ is also $G$-invariant, i.e. $\rho_f(R_gx) = \rho_f(x)$ for all $g \in G, x \in \mathcal{M}$.

We first recall the definition of $\rho_f$:

$$\rho_f(x) = \rho\left(f^{-1}(x)\right)\left|\det\frac{\partial f^{-1}(x)}{dx}\right| = \rho\left(f^{-1}(x)\right)\left|\det J_{f^{-1}}(x)\right|.$$

Since $f \in C^1(\mathcal{M}, \mathcal{M})$ is $G$-equivariant, we have $f \circ R_g = R_g \circ f$ for any $g \in G$. Also, since $\rho$ is $G$-invariant, we have $\rho \circ R_g = \rho$. Combining these properties, we see that:

$$\rho_f(R_g x) = \rho_f(R_g x) \frac{|\det J_{R_g}(x)|}{|\det J_{R_g}(x)|} = \frac{\rho_{R_{g^{-1}} \circ f}(x)}{|\det J_{R_g}(x)|} \qquad \text{(expanding definition of } \rho_f)$$

$$= \frac{\rho_{f \circ R_{g^{-1}}}(x)}{|\det J_{R_g}(x)|} = \rho\left((R_g \circ f^{-1})(x)\right) \frac{|\det J_{R_g \circ f^{-1}}(x)|}{|\det J_{R_g}(x)|} \qquad \text{(G-equivariance of } f)$$

$$= (\rho \circ R_g \circ f^{-1})(x) \frac{|\det J_{R_g}(f^{-1}(x)) J_{f^{-1}}(x)|}{|\det J_{R_g}(x)|} \qquad \text{(expanding Jacobian)}$$

$$= (\rho \circ f^{-1})(x) \frac{|\det J_{R_g}(f^{-1}(x))||\det J_{f^{-1}}(x)|}{|\det J_{R_g}(x)|} \qquad \text{(G-invariance of } \rho)$$

$$= \rho(f^{-1}(x))|\det J_{f^{-1}}(x)| \cdot \frac{|\det J_{R_g}(f^{-1}(x))|}{|\det J_{R_g}(x)|} \qquad \text{(rearrangement)}$$

$$= \rho_f(x) \cdot \frac{|\det J_{R_g}(f^{-1}(x))|}{|\det J_{R_g}(x)|} \qquad \text{(expanding definition of } \rho_f)$$

Now note that $G$ is contained in the isometry group, and thus $R_g$ is an isometry. This means $|\det J_{R_g}(x)| = 1$ for any $x \in \mathcal{M}$, so the right-hand side above is simply $\rho_f(x)$, which proves the theorem. $\qquad\square$

### A.4. Proof of Theorem 4

**Theorem 4.** *Let $(\mathcal{M}, h)$ be a closed Riemannian manifold. Let $\rho$ be a distribution over said manifold, and let $D_{KL}(\rho||\pi)$ be the Kullback–Leibler divergence between distributions $\rho$ and $\pi$. If we choose a $g : \mathcal{M} \to \mathbb{R}$ such that:*

$$g(x) = \log\left(\frac{\pi(x)}{\rho(x)}\right)$$

*for $x \in \mathcal{M}$, we have:*

$$\frac{\partial}{\partial t} D_{KL}(\rho||\pi) = -\int \rho \exp(g)||\nabla g||^2 \, dx$$

*Proof.* We start by noting the following by the Fokker-Planck equation:

$$\frac{\partial \rho}{\partial t} = \nabla \cdot (\rho \nabla g).$$

This gives:

$$\frac{\partial}{\partial t} D_{KL}(\rho||\pi) = \int \frac{\pi}{\rho} \frac{\partial \rho}{\partial t} \, dx$$

$$= \int \frac{\pi}{\rho} \nabla \cdot (\rho \nabla g) \, dx$$

$$= \int \left(\nabla \cdot \left(\frac{\pi}{\rho}(\rho \nabla g)\right) - (\rho \nabla g) \cdot \nabla \cdot \frac{\pi}{\rho}\right) \, dx$$

$$= \int \left(\nabla \cdot (\pi \nabla g) - (\rho \nabla g) \cdot \nabla \frac{\pi}{\rho}\right) \, dx$$

$$= -\int (\rho \nabla g) \cdot \nabla \frac{\pi}{\rho} \, dx,$$

where the final equality follows from the divergence theorem, since the integral of the divergence over a closed manifold is 0. Now if we choose $g$ such that:

$$g(x) = \log\left(\frac{\pi(x)}{\rho(x)}\right).$$

Then we have:

$$\frac{\partial}{\partial t} D_{KL}(\rho||\pi) = -\int (\rho \nabla g) \cdot \nabla \exp(g) \, dx$$

$$= -\int \rho \exp(g) ||\nabla g||^2 \, dx,$$

as desired.

$\square$

### A.5. Conjugation by $SU(n)$ as an Isometry

We now prove a lemma that shows that the group action of conjugation by $SU(n)$ is an isometry subgroup. This implies that Theorems 1 through 3 above can be specialized to the setting of $SU(n)$.

**Lemma 2.** *Let $G$ be the group action of conjugation by $SU(n)$, and let each $R_g$ represent the corresponding action of conjugation by $g \in SU(n)$. Then $G$ is an isometry subgroup.*

*Proof.* We first show that the matrix conjugation action of $SU(n)$ is unitary. For $R, X \in SU(n)$, note that the action of conjugation is given by $\text{vec}(RXR^{-1}) = (R^{-T} \otimes R)\text{vec}(X)$. We have that $R^{-T} \otimes R$ is unitary because:

$$(R^{-T} \otimes R)^*(R^{-T} \otimes R)$$
$$= (\overline{R^{-1}} \otimes R^*)(R^{-T} \otimes R) \qquad \text{(conjugate transposes distribute over } \otimes)$$
$$= (\overline{R^{-1}R^{-T}}) \otimes (R^*R) \qquad \text{(mixed-product property of } \otimes)$$
$$= (R^T R^{-T}) \otimes (I) = (I) \otimes (I) = I_{n^2 \times n^2} \qquad \text{(simplification)}$$

Now choose an orthonormal frame $X_1, \ldots, X_m$ of $T\mathcal{M}$. Note that $T\mathcal{M}$ locally consists of $SU(n)$ shifts of the algebra, which itself consists of traceless skew-Hermitian matrices (Gallier & Quaintance, 2020). We show $G$ is an isometry subgroup by noting that when it acts on the frame, the resulting frame is orthonormal. Let $g \in G$, and consider the result of action of $g$ on the frame, namely $R_g X_1, \ldots, R_g X_m$. Then we have:

$$(R_g X_i)^*(R_g X_j) = X_i^* R_g^* R_g X_j = X_i^* X_j.$$

Note for $i \neq j$, we have $X_i^* X_j = 0$ and for $i = j$ we see $X_i^* X_i = 1$. Hence the resulting frame is orthonormal and $G$ is an isometry subgroup. $\square$

## B. Manifold Details for the Special Unitary Group $SU(n)$

In this section, we give a basic introduction to the special unitary group $SU(n)$ and relevant properties.

**Definition.** The special unitary group $SU(n)$ consists of all $n$-by-$n$ unitary matrices $U$ (i.e. $U^*U = UU^* = 1$ for $U^*$ the conjugate transpose of $U$) that have determinant $\det(U) = 1$.

Note that $SU(n)$ is a smooth manifold; in particular, it has Lie structure (Gallier & Quaintance, 2020). Moreover, the tangent space at the identity (i.e. the Lie algebra) consists of traceless skew-Hermitian matrices (Gallier & Quaintance, 2020). The Riemannian metric is $\text{tr}(A^\top B)$.

### B.1. Haar Measure on $SU(n)$

**Haar Measure.** Haar measures are generic constructs of measures on topological groups $G$ that are invariant under group operation. For example, the Lie group $G = SU(n)$ has Haar measure $\mu_H : SU(n) \to \mathbb{R}$, which is defined as the unique measure such that for any $U \in SU(n)$, we have

$$\mu_H(VU) = \mu_H(UW) = \mu_H(U)$$

for all $V, W \in SU(n)$ and $\mu_H(G) = 1$.

A topological group $G$ together with its unique Haar measure defines a probability space on the group. This gives one natural way of defining probability distributions on the group, explaining its importance in our construction of probability distributions on Lie groups, specifically $SU(n)$.

To make the above Haar measure definition more concrete, we note from Bump (2004, Proposition 18.4) that we can transform an integral over $SU(n)$ with respect to the Haar measure into integrating over the corresponding diagonal matrices under eigendecomposition:

$$\int\limits_{SU(n)} f d\mu_H = \frac{1}{n!} \int\limits_T f(\text{diag}(\lambda_1, \ldots, \lambda_n)) \prod_{i<j} |\lambda_i - \lambda_j| d\lambda.$$

Thus, we can think of the Haar measure as inducing the change of variables with volume element

$$\text{Haar}(x) = \prod_{i<j} |\lambda_i(x) - \lambda_j(x)|^2.$$

To sample uniformly from the Haar measure, we just need to ensure that we are sampling each $x \in SU(n)$ with probability proportional to $\text{Haar}(x)$.

**Sampling from the Haar Prior.** We use Algorithm 1 (Mezzadri, 2007) for generating a sample uniformly from the Haar prior on $SU(n)$:

---
**Algorithm 1** Sampling from the Haar Prior on $SU(n)$
---
Sample $Z \in \mathbb{C}^{n \times n}$ where each entry $Z_{ij} = Z_{ij}^{(1)} + i Z_{ij}^{(2)}$ for independent random variables $Z_{ij}^{(1)}, Z_{ij}^{(2)} \sim \mathcal{N}(0, 1/2)$.
Let $Z = QR$ be the QR Factorization of $Z$.
Let $\Lambda = \text{diag}(\frac{R_{11}}{|R_{11}|}, \ldots, \frac{R_{nn}}{|R_{nn}|})$.
Output $Q' = Q\Lambda$ as distributed with Haar measure.

---

**B.2. Eigendecomposition on $SU(n)$**

One main step in the invariant potential computation for $SU(n)$ is to derive formulas for the eigendecomposition of $U \in SU(n)$ as well as formulas for differentiation through the eigendecomposition (recall that we must differentiate the $SU(n)$-invariant potential $f$ to get $SU(n)$-equivariant vector field $\nabla f$, as described in Section 5.2). This section first derives general formulas for how to do this for $U \in SU(n)$. In practice, such general methods often introduce instability, and thus, for the oft-used special cases of $n = 2, 3$, we derive explicit formulas for the eigenvalues based on finding roots of the characteristic polynomials (given by root formulas for quadratic/cubic equations).

B.2.1. DERIVATIONS FOR THE GENERAL CASE $SU(N)$

Here we reconstruct the steps of differentiation through eigendecomposition from Boyda et al. (2020, Appendix C) that allow efficient computation in our use-case. For our matrix-conjugation-invariant $SU(n)$ flow, we need only differentiate the eigenvalues with respect to the input $U \in SU(n)$.

For an input $U \in SU(n)$, let its eigendecomposition be $U = PDP^*$, where $w = \text{diag}(D) \in \mathbb{C}^n$ contains its eigenvalues, and $P = \begin{bmatrix} p_1 & \cdots & p_n \end{bmatrix} \in \mathbb{C}^{n \times n}$ with $p_i \in \mathbb{C}^n$ as its eigenvectors. Let $L$ denote our loss function, and write the downstream gradients in row vector format:

$$g = \begin{bmatrix} \frac{\partial L}{\partial \text{Re} w} & \frac{\partial L}{\partial \text{Im} w} \end{bmatrix} = \begin{bmatrix} g^{(1)} & g^{(2)} \end{bmatrix}.$$

Then following similar steps as in Boyda et al. (2020), we can compute the gradient of $L$ with respect to the real and imaginary parts of $U$ as follows:

$$\frac{\partial L}{\partial \text{Re} U} = \sum_{i=1}^n g_i^{(1)} \text{Re}\left(\overline{p_i} p_i^\top\right) + \sum_{i=1}^n g_i^{(2)} \text{Im}\left(\overline{p_i} p_i^\top\right)$$

$$\frac{\partial L}{\partial \text{Im} U} = -\sum_{i=1}^{n} g_i^{(1)} \text{Im}\left(\overline{p_i} p_i^\top\right) + \sum_{i=1}^{n} g_i^{(2)} \text{Re}\left(\overline{p_i} p_i^\top\right)$$

If we define

$$Q^{(1)} = \begin{bmatrix} g_1^{(1)} \overline{p_1} & \cdots & g_n^{(1)} \overline{p_n} \end{bmatrix} \qquad Q^{(2)} = \begin{bmatrix} g_1^{(2)} \overline{p_1} & \cdots & g_n^{(2)} \overline{p_n} \end{bmatrix}$$

Then we can write the gradients in terms of efficient matrix computations:

$$\frac{\partial L}{\partial \text{Re} U} = \text{Re}\left(Q^{(1)} P^\top\right) + \text{Im}\left(Q^{(2)} P^\top\right)$$

$$\frac{\partial L}{\partial \text{Im} U} = -\text{Im}\left(Q^{(1)} P^\top\right) + \text{Re}\left(Q^{(2)} P^\top\right).$$

### B.2.2. EXPLICIT FORMULA FOR $SU(2)$

We now derive an explicit eigenvalue formula for the $U \in SU(2)$ case. Let us denote $U = \begin{bmatrix} a + bi & -c + di \\ c + di & a - bi \end{bmatrix}$ for $a, b, c, d \in \mathbb{R}$ such that $a^2 + b^2 + c^2 + d^2 = 1$ as an element of $SU(2)$; then the characteristic polynomial of this matrix is given by

$$\det(\lambda I - U) = (\lambda - (a + bi))(\lambda - (a - bi)) + (c + di)(c - di) = (a - \lambda)^2 + b^2 + c^2 + d^2 = \lambda^2 - 2a\lambda + 1$$

and thus its eigenvalues are given by

$$\lambda_1 = a + i\sqrt{1 - a^2} = a + i\sqrt{b^2 + c^2 + d^2}$$

$$\lambda_2 = a - i\sqrt{1 - a^2} = a - i\sqrt{b^2 + c^2 + d^2}$$

**Remark.** We note that there is a natural isomorphism $\phi : S^3 \to SU(2)$, given by

$$\phi(a, b, c, d) = \begin{bmatrix} a + bi & -c + di \\ c + di & a - bi \end{bmatrix}$$

We can exploit this isomorphism by learning a flow over $S^3$ with a regular manifold flow like NMODE (Lou et al., 2020) and mapping it to a flow over $SU(2)$. This is also an acceptable way to obtain stable density learning over $SU(2)$.

### B.2.3. EXPLICIT FORMULA FOR $SU(3)$

We now derive an explicit eigenvalue formula for the $U \in SU(3)$ case. For the case of $U \in SU(3)$, we can compute the characteristic polynomial as

$$\det(\lambda I - U) = \det\left(\begin{bmatrix} \lambda - U_{11} & -U_{12} & -U_{13} \\ -U_{21} & \lambda - U_{22} & -U_{23} \\ -U_{31} & -U_{32} & \lambda - U_{33} \end{bmatrix}\right)$$
$$= \lambda^3 + c_2\lambda^2 + c_1\lambda + c_0$$

where

$$c_2 = -(U_{11} + U_{22} + U_{33})$$

$$c_1 = U_{11}U_{22} + U_{22}U_{33} + U_{33}U_{11} - U_{12}U_{21} - U_{23}U_{32} - U_{13}U_{31}$$

$$c_0 = -(U_{12}U_{23}U_{31} + U_{13}U_{21}U_{32} + U_{11}U_{22}U_{33} - U_{12}U_{21}U_{33} - U_{13}U_{31}U_{22} - U_{23}U_{32}U_{11})$$

Now to solve the equation

$$\lambda^3 + c_2\lambda^2 + c_1\lambda + c_0 = 0$$

we first transform it into a depressed cubic

$$t^3 + pt + q = 0$$

where we make the transformation

$$t = x + \frac{c_2}{3}$$

$$p = \frac{3c_1 - c_2^2}{3}$$

$$q = \frac{2c_2^3 - 9c_2 c_1 + 27 c_0}{27}$$

Now from Cardano's formula, we have the cubic roots of the depressed cubic given by

$$\lambda_{1,2,3} = \sqrt[3]{-\frac{q}{2} + \sqrt{\frac{q^2}{4} + \frac{p^3}{27}}} + \sqrt[3]{-\frac{q}{2} - \sqrt{\frac{q^2}{4} + \frac{p^3}{27}}}$$

where the two cubic roots in the above equation are picked such that they multiply to $-\frac{p}{3}$.

## C. Experimental Details for Learning Equivariant Flows on $SU(n)$

This section presents some additional details regarding the experiments that learn invariant densities on $SU(n)$ in Section 6.

### C.1. Training Details

Our DeepSet network (Zaheer et al., 2017) consists of a feature extractor and regressor. The feature extractor is a 1-layer tanh network with 32 hidden channels. We concatenate the time component to the sum component of the feature extractor before feeding the resulting 33 size tensor into a 1-layer tanh regressor network.

To train our flows, we minimize the KL divergence between our model distribution and the target distribution (Papamakarios et al., 2019), as is done in Boyda et al. (2020). In a training iteration, we draw a batch of samples uniformly from $SU(n)$, map them through our flow, and compute the gradients with respect to the batch KL divergence between our model probabilities and the target density probabilities. We use the Adam stochastic optimizer for gradient-based optimization (Kingma & Ba, 2015). The graph shown in Figure 1 was trained for 300 iterations with a batch size of 8192 and weight decay setting of 0.01; the starting learning rate for Adam was 0.01, and a multi-step learning rate schedule that decreased the learning rate by a factor of 10 every 100 epochs was used. We use PyTorch to implement our models and run experiments (Paszke et al., 2019). Experiments are run on one CPU and/or GPU at a time, where we use one NVIDIA RTX 2080Ti GPU with 11 GB of GPU RAM.

### C.2. Conjugation-Invariant Target Distributions

Boyda et al. (2020) defined a family of matrix-conjugation-invariant densities on $SU(n)$ as:

$$p_{toy}(U) = \frac{1}{Z} e^{\frac{\beta}{n} \operatorname{Re} \operatorname{tr}\left(\sum_k c_k U^k\right)},$$

which is parameterized by scalars $c_k$ and $\beta$. The normalizing constant $Z$ is chosen to ensure that $p_{toy}$ is a valid probability density with respect to the Haar measure.

More specifically, the experiments of Boyda et al. (2020) focus on learning to sample from the distribution with the above density with three components, in the following form:

$$p_{toy}(U) = \frac{1}{Z} e^{\frac{\beta}{n} \operatorname{Re} \operatorname{tr}\left(c_1 U + c_2 U^2 + c_3 U^3\right)}$$

We tested on three instances of the density, also used in Boyda et al. (2020):

| set $i$ | $c_1$ | $c_2$ | $c_3$ | $\beta$ |
|---------|-------|-------|-------|---------|
| 1 | 0.98 | -0.63 | -0.21 | 9 |
| 2 | 0.17 | -0.65 | 1.22 | 9 |
| 3 | 1 | 0 | 0 | 9 |

*Table 1.* Sets of parameters $c_1, c_2, c_3$ and $\beta$ used in the $SU(2)$ and $SU(3)$ experiments

Note that the rows of Figure 1 correspond to coefficient sets $3, 2, 1$, given in order from top to bottom.

### C.2.1. CASE FOR $SU(2)$

In the case of $n = 2$, we can represent the eigenvalues of a matrix $U \in SU(2)$ in the form $e^{i\theta}, e^{-i\theta}$ for some angle $\theta \in [0, \pi]$. We then have $\operatorname{tr}(U) = e^{i\theta} + e^{-i\theta} = 2\cos(\theta)$, so above density takes the form:

$$p_{toy}(U) = \frac{1}{Z} e^{c_1 \beta \cos\theta} \cdot e^{c_2 \beta \cos(2\theta)} \cdot e^{c_3 \beta \cos(3\theta)}.$$

### C.2.2. CASE FOR $SU(3)$

In the case of $n = 3$, we can represent the eigenvalues of $U \in SU(3)$ in the form $e^{i\theta_1}, e^{i\theta_2}, e^{i(-\theta_1 - \theta_2)}$. Thus, we have

$$\operatorname{Re} \operatorname{tr}(U) = \frac{1}{3} \Big( \cos(\theta_1) + \cos(\theta_2) + \cos(-\theta_1 - \theta_2) \Big)$$

and thus

$$p_{toy}(U) = \frac{1}{Z} e^{\frac{c_1 \beta}{3} \big( \cos(\theta_1) + \cos(\theta_2) + \cos(-\theta_1 - \theta_2) \big)}$$
$$\cdot e^{\frac{c_2 \beta}{3} \big( \cos(2\theta_1) + \cos(2\theta_2) + \cos(-2\theta_1 - 2\theta_2) \big)}$$
$$\cdot e^{\frac{c_3 \beta}{3} \big( \cos(3\theta_1) + \cos(3\theta_2) + \cos(-3\theta_1 - 3\theta_2) \big)}$$

## D. Sphere Isotropy Experiments

In this section, we illustrate the generality of the framework in the paper by presenting an additional invariant potential construction on a different manifold: the $n$-sphere $S^n$. Moreover, to demonstrate the need for enforcing equivariance of flow models, we directly compare our flow construction with a general purpose flow while learning a density with an inherent symmetry. The densities we decided to use for this purpose are sphere densities that are invariant to action by the isotropy group. Our model is able to learn these densities much better than previous manifold ODE models that do not enforce equivariance of flows (Lou et al., 2020), thus showing the ability of our model to leverage the desired symmetries. In fact, even on simple isotropy-invariant densities, our model succeeds while the free model without equivariance fails.

**Definition.** The unit $n$-sphere $S^n$ can be thought of as an embedded manifold of $\mathbb{R}^{n+1}$, given by

$$S^n = \{x \in \mathbb{R}^{n+1} : x_1^2 + \ldots + x_{n+1}^2 = 1\}$$

The discussions below will focus on the special case of the unit sphere $S^2$ in 3 dimensions.

### D.1. Isotropy Invariance on $S^2$

**Isotropy Group.** The isotropy group for a point $v \in S^2$ is defined as the subgroup of the isometry group which fixes $v$, i.e. the set of rotations around an axis that passes through $v$. In practice, we let $v = (0, 0, 1)$, so the isotropy group is the group of rotations on the $xy$-plane. An isotropy invariant density would be invariant to such rotations, and hence would look like a horizontally-striped density on the sphere.

**Invariant Potential Parameterization.** We design an invariant potential by applying a neural network to the free parameter. In the case of our specific isotropy group listed above, the free parameter is the $z$-coordinate. The invariant potential is simply a 1-input neural network on the $z$-coordinate of the input. As a result of this design, we see that the only variance in the learned distribution that uses this potential will be along the $z$-axis, as desired.

**Prior Distributions.** For proper learning with a normalizing flow, we need a prior distribution on the sphere that respects the isotropy invariance. There are many isotropy invariant potentials on the sphere. Natural choices include the uniform density (which is invariant to all rotations) and the wrapped distribution with the center at $v$ (Skopek et al., 2019; Nagano et al., 2019). For our experiments, we use the uniform density.

## D.2. Experiments

In this section, we present experiments on learning isotropy-invariant densities on the sphere. The specific density that we would like to learn is illustrated in Figure 2a, which is invariant under the isotropy group of rotations on the $xy$-plane.

We try to learn this density using our equivariant flow construction model (result in Figure 2b), and compare it to the previous manifold ODE model that do not enforce equivariance of flows in Lou et al. (2020) (result in Figure 2c). Both models are trained for 100 epochs with a learning rate of 0.001 and a batch size of 200.

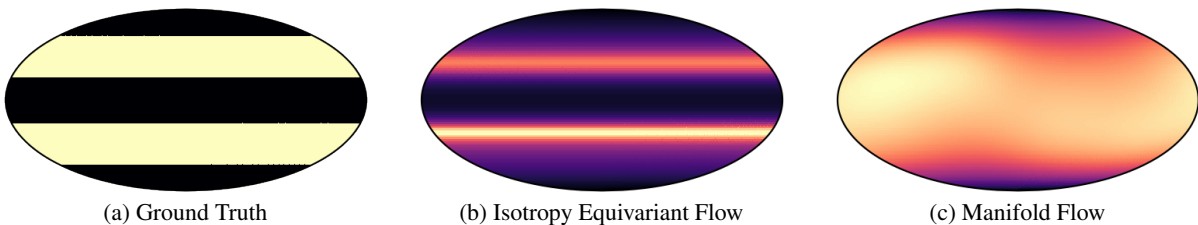

(a) Ground Truth       (b) Isotropy Equivariant Flow       (c) Manifold Flow

*Figure 2.* We compare the equivariant manifold flow and regular manifold flow on an invariant dataset. Note that our model is able to accurately capture the ground truth data distribution while the regular manifold flow struggles.

Despite our equivariant flow having fewer parameters (as both flows have the same width and the equivariant flow has an input dimension of 1), our model is able to capture the distribution much better than the base manifold flow. We believe this is due to the inductive bias of our equivariant model, which explicitly leverages the underlying symmetry.