# OpenReview forum: "Equivariant Manifold Flows"
_ICML.cc/2021/Workshop/INNF — INNF+ 2021 poster_

### Official Review · Reviewer_gZHF · 2021-06-10

**Rating:** Accept
**Confidence:** 1

**Summary:**


This paper presents several analysis results towards the goal of constructing equivariant normalizing flows, capable of incorporating symmetries to learn and represent probabilities densities that respect a broad class of invariances. The main contributes are Theorems 1-3 and how they're used to construct equivariant flows in Sec 5. Experimental results of using the method to learn densities on SU(n) invariant to conjugation are illustrated that show the method learning to approximate SU(2) and SU(3) densities.

It's not clear whether the results are more showing that existing methods can be used to construct equivariant manifold flows, or whether its more that new flow architectures were constructed to satisfy properties that others did not satisfy. Sec 4.2 could be improved by being clearer about this. My understanding is that it's more the former -- essentially it shows existing methods can be used to satisfy the Thms 1-3.

The paper is missing quantification of some claims about the performance of the results. Fig 1a: "Our model has better behavior in low-density regions, and more smoothly captures the targets in high-density regions"; Fig 1b. "Our model better captures the geometry of the target densities." Qualitatively these seems true -- but the paper would be strengthened by some sort of quantification of these claims.

[A] has come out since the INNF deadline, although the submission authors are probably already aware, the paper should discuss and possibly compare to [A]. The submitted paper seems potentially more general (L28 claims applicability to arbitrary Riemannian manifolds).

***I do not have the expertise to provide a more comprehensive review than this; in particular, I do not think I can, within a reasonable amount of time, verify the claims of Theorems 1-4, which seem central to the paper.

[A] E(n) Equivariant Normalizing Flows https://arxiv.org/pdf/2105.09016.pdf


**Justification For Rating:**

From my incomplete understanding, the work seems novel, and the experiments seem promising, although somewhat preliminary.

---

### Official Review · Reviewer_837x · 2021-06-12

**Rating:** Borderline Accept
**Confidence:** 3

**Summary:**

This paper lays the foundation of learning flows that induce symmetry-invariant distributions. The paper connects invariant densities with equivariant flows, demonstrated how to use manifold ODEs to construct equivariant manifold flows, and proves the flows induced by invariant potentials suffice to learn any smooth invariant distribution.
Then the authors verify the learned densities on SU(n) empirically.

**Justification For Rating:**

Pros:
The paper does clarify the relationship between invariant manifold densities and equivariant flows, even though I find most theorems straightforward and should be referred to as claims.
The paper yields some moderate contribution in sampling and learning from equivariant manifold flows.

Cons:
From the experiments, it's hard to tell whether the proposal is much better than Boyda et al. Boyda et al show better performance when the density is higher while the newly proposed method shows better performance at low-density regions.


typo: "over said manifold" in Thm 4

I believe the contributions are sufficient for the workshop.

---

### Official Review · Program_Chairs · 2021-06-14

**Rating:** Borderline Accept
**Confidence:** 5

**Summary:**

The goal of this work is to learn a density on manifolds that is invariant with respect to the action of a symmetry group. It introduces Monge-Ampere-type ODE-flows on manifolds where the potentials are invariant with respect to the action of a symmetry group which results in equivariant ODE-flows.

**Justification For Rating:**

General comments:

The main idea and derivations are sound, the text is clear and the problem (learning invariant densities) is very important.

The proposed model is at the intersection of 3 existing ideas: 1) ODE flows on manifolds [6, 7]; 2) ODE flows with gradient fields [3] and 3) That gradient fields of invariant potentials are equivariant [1,2,4]. Novelty-wise, it is a good contribution but not very strong on its own.

Experiments show promising results, but analysis is quite limited. It would substantially strengthen the paper to have a more quantitative and detailed analysis. For example, one could look more closely at the tails of the learned density in Fig.1 and provide a more quantitative comparison between [5] and this work.
A discussion of the computational complexity of using ODE flows versus spline flows in [5] would also be very relevant.

Specific comments:

Related work. The section "Equivariant machine learning" could also cite [1] as concurrent work to [4].

The idea of ODE flows which are gradient fields of a scalar potential was explored in [3], so it would be good to mention in the related work.

The theorems are quite simple and have been addressed in previous work on flows:
- Theorem 1 ---> Has also been proven in [2] (lemma 2, page 42) and in [4]
- Theorem 2 ---> straightforward
- Theorem 3 ---> Has also been proven in [2] (lemma 1, page 42)

Maybe they can be absorbed in the main text or "downgraded" to lemmas?


[1] Rezende, D.J., Racanière, S., Higgins, I. and Toth, P., 2019. Equivariant hamiltonian flows. arXiv preprint arXiv:1909.13739.

[2] Papamakarios, G., Nalisnick, E., Rezende, D.J., Mohamed, S. and Lakshminarayanan, B., 2019. Normalizing flows for probabilistic modeling and inference. arXiv preprint arXiv:1912.02762.

[3] Zhang, L. and Wang, L., 2018. Monge-amp\ere flow for generative modeling. arXiv preprint arXiv:1809.10188.

[4] Köhler, J., Klein, L. and Noé, F., 2020, November. Equivariant flows: exact likelihood generative learning for symmetric densities. In International Conference on Machine Learning (pp. 5361-5370). PMLR.

[5] Boyda, D., Kanwar, G., Racanière, S., Rezende, D.J., Albergo, M.S., Cranmer, K., Hackett, D.C. and Shanahan, P.E., 2021. Sampling using SU (N) gauge equivariant flows. Physical Review D, 103(7), p.074504.

[6] Lou, A., Lim, D., Katsman, I., Huang, L., Jiang, Q., Lim, S. N., and De Sa, C. M. Neural manifold ordinary differential equations. In Advances in Neural Information Processing Systems, volume 33, pp. 17548–17558, 2020.

[7] Mathieu, E. and Nickel, M. Riemannian continuous normalizing flows. In Advances in Neural Information Processing Systems, volume 33, pp. 2503–2515, 2020.

---

### Decision · Program_Chairs · 2021-06-14

**Decision:**

Accept (poster)

**Comment:**

The paper's topic is in line with the theme of the workshop, and the reviews were fairly positive. We have therefore decided to accept this submission to the workshop. For the camera ready version, please take into account the suggestions by the reviewers, in particular the suggestion to change theorems to lemmas or claims in the text. Note that you do not have to compare to work that came out after the INNF submission deadline, although you are free to discuss it as concurrent work if you would like to.